# The Role of NT-proBNP Levels in the Diagnosis of Hypertensive Heart Disease

**DOI:** 10.3390/diagnostics15010113

**Published:** 2025-01-06

**Authors:** Angeliki Mouzarou, Nikoleta Hadjigeorgiou, Despo Melanarkiti, Theodora Eleni Plakomyti

**Affiliations:** Department of Cardiology, Paphos General Hospital, State Health Organization Services, Paphos 8026, Cyprus

**Keywords:** hypertension, hypertensive heart disease, NT-proBNP, natriuretic peptide, diastolic heart failure

## Abstract

Hypertension is a major risk factor of various cardiac complications, including hypertensive heart disease (HHD). This condition can lead to a number of structural and functional changes in the heart, such as left ventricular hypertrophy, diastolic dysfunction, and, eventually, systolic dysfunction. In the management of hypertensive heart disease, early diagnosis and appropriate treatment are crucial for preventing the progression to congestive heart failure. One potential diagnostic marker that has gained attention in recent years is the N-terminal pro-brain natriuretic peptide (NT-proBNP). The natriuretic peptides, including the brain natriuretic peptide (BNP) and its inactive N-terminal fragment, are secreted by the myocardium in response to increased wall stress and volume overload. In patients with hypertensive heart disease, increased NT-proBNP levels may reflect the structural and functional changes occurring in the myocardium as a result of chronic pressure overload. Several studies have investigated the diagnostic utility of NT-proBNP in hypertensive heart disease. NT-proBNP levels can be a useful adjunct in the diagnosis of hypertensive heart disease, particularly in the assessment of diastolic dysfunction and left ventricular hypertrophy. This review paper explores the role of NT-proBNP levels in the diagnosis of hypertensive heart disease.

## 1. Introduction

Hypertension is a major risk factor of various cardiac complications, including hypertensive heart disease (HHD) [1]. This condition can lead to a number of structural and functional changes in the heart, such as left ventricular hypertrophy, diastolic dysfunction, and, eventually, systolic dysfunction [2]. In the management of hypertensive heart disease, early diagnosis and appropriate treatment are crucial for preventing the progression to congestive heart failure. One potential diagnostic marker that has gained attention in recent years is the N-terminal pro-brain natriuretic peptide (NT-proBNP). The purpose of this paper is to explore the role of NT-proBNP levels in the diagnosis of hypertensive heart disease. The natriuretic peptides, including the brain natriuretic peptide (BNP) and its inactive N-terminal fragment, are secreted by the myocardium in response to increased wall stress and volume overload [3]. Elevated levels of these peptides have been associated with various cardiac conditions, including systolic and diastolic heart failure [3,4]. In patients with hypertensive heart disease, increased NT-proBNP levels may reflect the structural and functional changes occurring in the myocardium as a result of chronic pressure overload. Several studies have investigated the diagnostic utility of NT-proBNP in hypertensive heart disease. A study showed that NT-proBNP levels were significantly higher in patients with left ventricular hypertrophy compared to those without [5]. Moreover, NT-proBNP levels correlated with the severity of diastolic dysfunction, with higher levels observed in patients with a restrictive filling pattern [3]. NT-proBNP has been found to be superior to BNP (the originally used biomarker) in the detection of left ventricular systolic dysfunction in the community setting [2]. This is likely due to the longer half-life and higher plasma concentrations of NT-proBNP, which may provide a more sensitive and reliable marker of myocardial stretch and remodeling. In conclusion, NT-proBNP levels can be a useful adjunct in the diagnosis of hypertensive heart disease, particularly in the assessment of diastolic dysfunction and left ventricular hypertrophy. Further research is warranted in order to establish the optimal cut-off values and the incorporation of NT-proBNP into clinical decision-making algorithms for the management of hypertensive heart disease (Table 1).

## 2. Pathophysiology of Hypertensive Heart Disease (HHD)

Chronic arterial hypertension disrupts myocardial structure and function through mechanisms like hypertrophy, fibrosis, and ischemia. This can lead to systolic and diastolic dysfunction, atrial dilation, and structural changes in the coronary arteries. Left ventricular hypertrophy (LVH) is a hallmark, driven by both hemodynamic and non-hemodynamic factors [6]. Hemodynamic stress, as described by Laplace’s law, results in increased wall stress and oxygen demand, triggering gene expression changes that lead to myocyte hypertrophy and extracellular matrix alterations. From a pathophysiological and anatomical perspective, hypertensive heart disease (HHD) can manifest in four distinct patterns of hypertrophy depending on left ventricular (LV) dilation: (1) Eccentric Non-Dilated Hypertrophy features the thickening of the LV walls with a normal chamber size and a relative wall thickness (RWT) ≤ 0.42; (2) Eccentric Dilated Hypertrophy involves both the hypertrophy and enlargement of the LV chamber, resulting from chronic volume overload; (3) Concentric Non-Dilated Hypertrophy is characterized by thickened LV walls without chamber dilation and a RWT > 0.42; and (4) Concentric Dilated Hypertrophy combines the increased wall thickness with chamber enlargement, reflecting the prolonged pressure overload that eventually causes both myocardial thickening and dilation [7]. These patterns illustrate the complex adaptive response of the heart to chronic hypertension. Non-hemodynamic factors, such as the renin–angiotensin–aldosterone system, demographic factors, such as gender and ethnicity, obesity, and genetics, play a significant role in myocardial hypertrophy. It is worth noting that it is not fully understood why patients with HHD develop a specific LVH pattern.

Myocardial fibrosis is a key component of HHD [8]. It can be classified as either interstitial (also known as reactive or diffuse) or replacement (also known as reparative) fibrosis. Interstitial fibrosis involves the accumulation of fibrous tissue in interstitial and perivascular spaces without significant myocardial cell loss. In contrast, replacement fibrosis arises from the loss of myocardial cells, which are replaced by scar tissue following events like myocardial infarction [9]. The relationship between these fibrosis types is not fully understood, as they may occur concurrently [6]. The development of myocardial fibrosis is influenced by both hemodynamic and non-hemodynamic factors. Hemodynamic stress, such as chronic pressure overload, triggers increased collagen production (types I and III) in a reparative response. Non-hemodynamic factors, particularly the renin–angiotensin–aldosterone system, also contribute to fibrosis by disrupting the balance of profibrotic and antifibrotic molecules through pathways activated by angiotensin II and aldosterone. This leads to fibrosis in the myocardium of both ventricles and the left atrium, with significant changes in the extracellular matrix composition.

HHD is associated with a progressive transition to heart failure. In this process, myocardial hypertrophy and fibrosis contribute to myocardial wall stiffness and the diastolic dysfunction of the left ventricle. Eccentric hypertrophy is typically associated with heart failure with a reduced ejection fraction (HFrEF), while concentric hypertrophy, particularly after events like myocardial infarction, is more likely to result in heart failure with a preserved ejection fraction (HFpEF). Additionally, changes in the left atrium’s morphology begin early, even when the LV appears normal [10,11]. Microstructural changes often precede macroscopic alterations, resulting in the functional impairment of the atrium despite its normal size. Left atrial remodeling is driven by both hemodynamic factors, such as an increased afterload, leading to elevated filling pressures and wall tension, and non-hemodynamic factors, including neurohormonal activation [11]. The latter involves the renin–angiotensin–aldosterone system, natriuretic peptides (NPs), and endothelin-1, which contribute to inflammation, fibrosis, and further atrial remodeling. It is worth mentioning that several mechanisms have been proposed to explain how systemic hypertension leads to the remodeling of the right ventricle, with fewer identified for the right atrium, although the process is not yet fully understood [10].

Moreover, increased sympathetic nervous system activity, driven by neurohormonal activation and involving norepinephrine, plays a pivotal role in the pathophysiology of HHD [12].

Ultimately, persistent arterial hypertension causes arteriosclerosis in systemic and coronary vessels and leads to microvascular rarefaction. This results in a mismatch between the coronary blood supply and myocardial oxygen demand, leading to myocardial ischemia [10].

## 3. Clinical Course of Hypertensive Heart Disease (HHD)

HHD is a complex and multifaceted condition that arises from the prolonged exposure of the cardiovascular system to elevated blood pressure. This chronic condition can lead to a range of structural and functional changes in the heart, ultimately culminating in the development of heart failure. 

Several risk factors such as increasing age, family history, high sodium diets (greater than 3 g/day), physical inactivity, and excessive alcohol consumption are strongly associated with the development of hypertension [13]. Common medical conditions in hypertensive individuals, such as diabetes mellitus, obesity, and coronary artery disease (CAD), can also influence the pattern of hypertrophic response [14]. In more detail, CAD seems to be associated with an increased LV diastolic dimension and a higher prevalence of eccentric LVH [15]. Diabetes mellitus has been associated with a concentric hypertrophic response, whereas obesity, characterized as a volume-overload state, has been associated predominantly with eccentric hypertrophy [16,17]. It is known that hypertension is more common in women and in certain ethnic groups such as African Americans, Caucasians, and Spanish people. The prevalence of hypertension in the African American population, at 45.0% for males and 46.3% for females, is the highest of any ethnic group in the world. In Caucasians, the rate is at 34.5% for males and 32.3% for females, whereas, in Spanish people, the rate is at 28.9% for males and 30.7% for females [18]. In addition to the highest rate of hypertension, black Americans have a higher risk of developing heart failure and higher average blood pressure which develops at an earlier age, and are less amenable to treatment [13].

The clinical course of HHD can be broadly divided into several stages. In the early stages, patients may be asymptomatic, with the only detectable changes being subtle alterations in cardiac structure and function. As the disease progresses, patients may begin to experience symptoms of heart failure, such as shortness of breath and fatigue, particularly with exertion. 

Over time, the structural and functional changes in the heart can lead to the development of evident heart failure, which can manifest as either HFpEF or HFrEF [19,20,21,22]. In HFpEF, the ventricle is unable to fill with blood efficiently due to reduced compliance, while, in HFrEF, the ventricle is unable to effectively eject blood due to impaired contractility [19,21,22]. The progression of HHD is influenced by a variety of factors, including the severity and duration of hypertension, the presence of comorbidities like diabetes or obesity, and the effectiveness of treatment interventions [19,20,21,22].

LVH can be regarded as a biological marker that reflects and integrates long-term exposure not only to pressure overload, but also to various hemodynamic and non-hemodynamic factors. All these factors may contribute to the progression and destabilization of atherosclerotic lesions, ultimately leading to adverse clinical events [23]. LVH can partially or totally regress following antihypertensive treatment and lifestyle changes including losing excessive weight and decreasing salt intake. Major determinants are treatment duration and the degree of blood pressure (BP) reduction, particularly the 24 h average BP [24]. Angiotensin II receptor blockers (ARBs) and angiotensin-converting enzyme inhibitors (ACE inhibitors) seem to be the most effective drugs for reversing LVH. Accumulated data from several studies show that the regression of LVH is associated with a significant reduction in the subsequent risk of cardiovascular disease [25,26].

The early recognition and management of HHD is essential for slowing disease progression and improving patient outcomes. The effective control of hypertension, through lifestyle modifications such as diet, exercise, and weight management, along with appropriate medications, plays a critical role in mitigating the advancement of HHD. Medications such as ACE inhibitors, ARBs, beta-blockers, diuretics, and calcium channel blockers are commonly used to manage blood pressure, prevent hypertrophy progression, and manage symptoms of heart failure and arrhythmias. Regular patient follow-up is essential in order to adjust treatment and manage complications [27].

An important biomarker that has garnered attention in the context of HHD is NT-proBNP. Elevated levels of NT-proBNP have been observed in patients with HHD, as the increased pressure and volume load on the ventricles lead to the release of this peptide [28]. Several studies have investigated the utility of NT-proBNP in the diagnosis, management, and prognostic assessment of patients with HHD.

### Synthesis and Role of Natriuretic Peptides

Brain natriuretic peptide (BNP) is a cardiac hormone composed of 32 amino acids. The BNP gene is situated on chromosome 1 and produces the prohormone known as proBNP. This prohormone is subsequently converted into the biologically active form BNP and the biologically inactive NT-proBNP. Data from in vitro experiments indicate that the proteolytic enzyme furin is responsible for splitting the prohormone into its two subsections [29]. The biologically active BNP and the remaining segment of the prohormone, NT-proBNP (comprising 76 amino acids), can be detected in human blood using immunoassay techniques. Cardiac myocytes are the primary producers of BNP-related peptides, though cardiac fibroblasts have also been found to produce BNP [29]. Despite their name, brain NPs are primarily expressed by ventricular cardiomyocytes in response to cardiac stress, such as volume overload. In addition to myocardial wall stress, factors like cardiomyocyte damage or hypoxia can also trigger NPs’ gene expression in the ventricular myocardium. Secreted NPs induce natriuresis and vasodilation, along with inhibiting the renin–angiotensin system and adrenergic activity [30,31]. The production of ventricular NT-proBNP is known to increase in cases of cardiac failure and is particularly elevated in areas affected by myocardial infarction [29].

BNP has a short half-life of around 20 min and is primarily cleared through natriuretic receptor internalization and enzymatic degradation by neutral endopeptidase in the bloodstream. In contrast, NT-proBNP, with a longer plasma half-life of approximately 90 min, is excreted by the kidneys [32]. Due to its longer half-life and higher plasma levels compared to BNP, NT-proBNP serves as a key biomarker in cardiology, especially for diagnosing and managing heart failure, making it the preferred marker in clinical practice [33].

Additionally, studies have shown that plasma NT-proBNP is a sensitive indicator of cardiac dysfunction, both with and without systolic dysfunction. Compared to BNP, NT-proBNP tended to be more accurate in identifying lesser degrees of LV-dysfunction. Furthermore, because of its good negative predictive value, NT-proBNP could be an easy and effective tool to rule out severe systolic LV-dysfunction in high risk patients [34].

Genes that elevate circulating BNP levels help protect against hypertension, heart structural remodeling, and metabolic diseases [35]. Beyond their therapeutic potential, the increased production and release of NPs in heart failure have led to their widespread use as diagnostic and prognostic biomarkers of HF. The superiority of BNP and NT-proBNP as biomarkers is attributed to both hemodynamic and non-hemodynamic mechanisms that regulate BNP gene expression. Non-hemodynamic factors influencing BNP gene expression include stimulation by hormones such as endothelin and angiotensin II, as well as cytokines and myocardial hypoxia.

## 4. NT-proBNP in Cardiovascular Disease

### 4.1. Heart Failure

Plasma concentrations of NT-proBNP are elevated regardless of the underlying cause of heart failure, whether it be ischemic, hypertensive, or valvular, or due to a primary cardiomyopathic process. The severity of heart failure correlates directly with higher NT-proBNP levels [36]. Consequently, elevated NT-proBNP levels are associated with the severity of left ventricular systolic dysfunction, right ventricular systolic dysfunction, left ventricular pressures, and alterations in left ventricular filling [30].

### 4.2. Acute Coronary Syndromes

While NT-proBNP is not directly used to diagnose CAD, elevated levels can indicate the presence of heart failure or significant ventricular dysfunction in patients with CAD. It also aids in risk stratification, identifying patients at a higher risk of adverse outcomes. Recent research highlights the high prognostic value of NT-proBNP in patients with acute coronary syndrome, showing that it serves as an independent predictor of the severity of CAD [37]. Following an acute myocardial infarction, NT-proBNP plasma concentrations rise, correlating with the severity of the infarction. Additionally, studies have found that NT-proBNP is a strong predictor of short-term outcomes in patients with acute ST-elevation myocardial infarction (STEMI), with higher levels significantly linked to an increased risk of complications such as arrhythmias, heart failure, and death within the first seven days post-infarction [38].

### 4.3. Arterial Hypertension

The plasma levels of NT-proBNP partially reflects a left ventricular myocardial mass, suggesting its potential as a marker for LV hypertrophy in evaluating patients with essential hypertension [39]. NT-proBNP levels are also being studied as a means to guide antihypertensive therapy. By monitoring NT-proBNP, clinicians can identify patients whose hypertension is causing subclinical cardiac stress, enabling more aggressive or tailored treatment approaches to preventing future cardiovascular events [40].

### 4.4. Valvular Heart Disease

NT-proBNP levels consistently increase with the severity of heart valve lesions, though the implications of elevated BNP or NT-proBNP vary by valve type. In aortic valve stenosis, higher levels correlate with stenosis severity, symptoms, and prognosis, and indicate left ventricular dysfunction, with poor outcomes in conservative treatment and an increased risk of postoperative complications. In mitral regurgitation, elevated NPs reflect the extent of regurgitation and subclinical left ventricular dysfunction. For mitral stenosis, serum levels correlate with stenosis severity and increased pulmonary pressures. Elevated serum levels of NPs are also observed in aortic valve regurgitation and mixed valvular disease, indicating general heart strain across different valve dysfunctions [41].

### 4.5. Atrial Fibrillation

NT-proBNP levels are shown to be increased in patients with atrial fibrillation (AF). Higher levels of NT-proBNP are observed in patients with permanent AF compared to those with paroxysmal AF, likely due to the greater and more sustained hemodynamic burden in permanent AF. The increase in NT-proBNP in AF patients is primarily driven by atrial and ventricular strain, as AF induces irregular atrial contractions, elevated wall stress, and neurohormonal activation, including the renin–angiotensin–aldosterone system and sympathetic nervous system [42]. Recent studies confirm that NT-proBNP levels can rise independently of anatomical changes, such as left atrial enlargement, and may reflect the subclinical atrial remodeling or fibrosis. Therefore, NT-proBNP serves as a robust biomarker and risk predictor of AF, offering valuable diagnostic and prognostic information in patients with or without structural cardiac abnormalities [43].

## 5. Diagnostic and Prognostic Value of NT-proBNP

Plasma NT-proBNP levels are a valuable diagnostic tool for heart failure, offering high sensitivity, specificity, and a strong positive predictive value. In clinical settings, Nt-proBNP testing is primarily used as a “rule out” test for suspected new cases of heart failure in patients presenting with breathlessness. However, it should not be seen as a substitute for an echocardiogram and a comprehensive cardiological assessment, which are necessary for patients with elevated NT-proBNP levels. For patients experiencing dyspnea, a low NT-proBNP concentration can effectively rule out decompensated heart failure. Conversely, a very high NT-proBNP concentration strongly supports a heart failure diagnosis but does not exclude the possibility of other conditions contributing to the symptoms [44].

While specific ‘target thresholds’ for NT-proBNP are not fully established, morbidity and mortality in chronic heart failure (CHF) significantly increase when NT-proBNP concentrations exceed 500 pg/mL. The repeated measurement of NP levels is essential for monitoring heart disease progression and assessing the effectiveness of medical therapy. For instance, fluctuations in NT-proBNP levels during a hospital stay has been identified as an independent predictor of hospital readmission within 6 months and mortality in patients admitted for decompensated heart failure [30]. Moreover, it has been identified as the most reliable predictor of adverse outcomes. Increasing levels of NT-proBNP during a hospital stay are associated with a poorer prognosis, whereas decreasing levels suggest a more favorable outlook. This dynamic measurement can provide critical insights into patient management and prognostication [44].

## 6. NT-proBNP in Clinical Practice

### 6.1. Guidelines and Recommendations

NT-proBNP is recommended for diagnosing heart failure in patients who present with symptoms like dyspnea. According to the European Society of Cardiology (ESC) guidelines, CHF is considered unlikely if BNP levels are below 100 pg/mL and NT-proBNP levels are below 400 pg/mL. Conversely, heart failure is deemed likely when BNP exceeds 400 pg/mL and NT-proBNP surpasses 2000 pg/mL. The diagnosis remains uncertain when BNP levels range between 100–400 pg/mL and NT-proBNP levels fall between 400–2000 pg/mL [45]. Unfortunately, there are still no clear cut-off points where we can differentiate between the stages of heart failure. According to a study, individuals without any history of HF but with BNP ≥ 100 pg/mL are at an equal or higher risk than those with a HF history whose BNP is <100 pg/m, so BNP or Nt-proBNP may be useful in identifying asymptomatic individuals at high risk for future cardiovascular events [46]. Furthermore, NT-proBNP levels can be monitored over time to assess the effectiveness of treatment and track disease progression. Generally, decreasing levels of NT-proBNP are indicative of a favorable response to treatment. The guidelines also emphasize the importance of considering individual patient factors such as age, renal function, and comorbidities when interpreting NT-proBNP levels, underscoring the need for a personalized approach in the management of heart failure [47].

### 6.2. Integration of NT-proBNP into Diagnostic and Therapeutic Algorithms

Integrating NT-proBNP into diagnostic and therapeutic algorithms for cardiovascular diseases (CVDs) significantly enhances clinical decision-making and patient outcomes. This biomarker can be used for early screening in at-risk populations, such as those with hypertension or diabetes, to detect cardiac changes before symptoms are apparent, allowing for earlier interventions [48]. As mentioned earlier, NT-proBNP is useful beyond diagnosing heart failure, in managing other CVDs, such as acute coronary syndromes and valvular heart disease, helping guide decisions on interventions such as surgical versus medical management [49]. For a comprehensive approach, NT-proBNP levels can be integrated into strategies that include other biomarkers like troponin or combined with echocardiographic and clinical data for a multidimensional assessment of cardiovascular health [50]. Incorporating NT-proBNP into clinical algorithms not only improves the precision of diagnostics and efficacy of treatments but also supports personalized care, crucial for managing complex cardiovascular conditions [51]. This approach aligns with modern medicine’s emphasis on tailored healthcare strategies based on individual patient profiles and dynamic clinical data [52].

### 6.3. Limitations and Considerations

Several confounding factors can influence plasma NT-proBNP levels, making its interpretation complex: (1) Sex: Women have higher NT-proBNP levels than men, which means increased levels may have a greater predictive value for adverse events in women [30,53]. (2) Race: NT-proBNP levels vary by race. For instance, African-American and Hispanic individuals often have higher levels than Caucasians within the same New York Heart Association (NYHA) class [18,30]. (3) Renal Insufficiency: This condition can raise NT-proBNP levels regardless of the presence of heart failure, as impaired kidney function slows the clearance of the peptide [30]. (4) Anemia: Patients with anemia generally exhibit higher NT-proBNP levels, possibly due to increased cardiac output or hypoxia [30,54]. (4) Cardio-Renal Syndrome: Significantly elevated NT-proBNP levels are also observed in patients with cardio-renal syndrome, where both the heart and kidney function is compromised [30,55]. (5) Obesity: Interestingly, obesity is associated with lower NT-proBNP levels, even when hypertension or left ventricular (LV) dysfunction is present [56]. The pathophysiological link between obesity and low BNP/NT-proBNP has not been fully elucidated yet, but several mechanisms have been proposed. Obesity leads to a decreased BNP concentration, mainly by enhancing its clearance through the increased concentration of natriuretic peptide receptor C (NPR-C) well as through increased renal filtration. Obesity also decreases BNP activity by decreasing NPR-A concentrations and BNP intracellular signaling pathways [31]. (6) Supraventricular Arrhythmia: Conditions like atrial fibrillation can increase NT-proBNP levels, suggesting that higher normal ranges may be necessary for diagnosing heart failure in these patients [30,57]. Understanding these factors is crucial for accurately assessing NT-proBNP levels and their implications in diagnosing and managing heart failure and related conditions.

## 7. Emerging Research and Future Directions

### 7.1. Novel Applications

Novel applications of NT-proBNP extends its use beyond traditional heart failure diagnosis and management. It is increasingly being utilized in various clinical scenarios, including risk stratification for surgeries, cardiotoxicity monitoring in cancer therapy, and the monitoring of chronic conditions like Chronic Kidney Disease (CKD) [58,59,60]. These expanding roles highlight NT-proBNP’s versatility as a biomarker, with potential applications across a broader range of cardiovascular and systemic diseases.

### 7.2. Technological Advances

Over the years, there has been significant advancement in the methods used to detect NT-proBNP, improving sensitivity, specificity, and usability in clinical practice. New high-sensitivity assays have been developed, capable of detecting lower serum concentrations of NT-proBNP. These assays provide better accuracy in diagnosing heart failure, especially in the early stages or in populations with typically lower NT-proBNP levels, such as younger patients. Advanced assays have also been designed to minimize cross-reactivity with other peptides and proteins, reducing the likelihood of false-positive or false-negative results [61,62]. Some newer techniques are focused on enabling the remote monitoring of NT-proBNP levels in patients with CHF. This includes wearable technology and home testing kits, allowing patients to regularly monitor their NT-proBNP levels and share the data with healthcare providers via digital platforms [63].

## 8. Future Studies

In the near future, algorithm building will take into consideration clinical and echocardiographic parameters as well as NP measurements, and this may better ensure the correct diagnosis and categorization of patients with worsening prognosis. A complete algorithm including clinical, echocardiographic, and laboratory examinations will lead to a better stratification in the setting of heart failure [64].

## 9. Hypertensive Heart Disease (HHD) and NT-proBNP

HHD involves a range of cardiac changes, and NT-proBNP is crucial for the early detection of these abnormalities. Key features linked to NT-proBNP include left atrial (LA) remodeling, where early microstructural changes occur before LA enlargement, and LVH, which reflects increased cardiac stress. NT-proBNP also correlates with both diastolic and systolic dysfunction, making it a crucial biomarker for identifying early signs of atrial and ventricular impairment in HHD, even before significant structural alterations become apparent (Table 2).

## 10. Atrial Remodeling

In early-stage HHD, even when the left ventricle (LV) remains normal-sized, the LA can undergo significant microstructural changes, such as fibrosis, before any noticeable dilatation becomes apparent. These microstructural alterations often precede macroscopic changes like LA dilatation, leading to functional impairment of the LA despite its normal size [73]. Elevated NT-proBNP levels are closely linked to LA remodeling, as studies show that higher NT-proBNP correlates with increased LA volume, reduced LA function, and diminished LA strain. These associations highlight that NT-proBNP can reflect changes in the LA independent of left ventricular parameters [71].

### 10.1. Macroscopic Changes: Left Atrial Diameter and Left Atrial Volume Index

LA enlargement is closely linked to chronic LA dysfunction and is a key indicator of the duration and severity of diastolic heart failure [68]. In HHD, a larger LA, as measured by the parasternal short-axis view, is associated with higher NT-proBNP levels [74]. An increase in the left atrial volume index (LAVI) also correlates with elevated NT-proBNP levels [75]. A stronger correlation is observed when the E/E’ ratio is high, indicating significant diastolic dysfunction. However, in patients with a lower E/E’ ratio, this correlation between LAVI and NT-proBNP is less pronounced. Instead, the systolic functional parameter S, which reflects the peak systolic tissue Doppler velocity of the mitral annulus, shows a stronger association with NT-proBNP levels. A low E/E’ ratio generally signifies preserved LV wall compliance, suggesting that elevated NT-proBNP levels in these cases may more accurately reflect changes in LV function rather than LA function alone [72]. The correlation between NT-proBNP levels and LA enlargement holds promise for identifying high-risk patients. Nonetheless, additional research is required to confirm and refine this potential diagnostic approach.

### 10.2. Microstructural Changes: Speckle Tracking of Left Atrium

Recent research suggests that LA dysfunction may precede detectable structural changes, such as LA dilatation [69]. Studies indicate that all three phases of LA function, reservoir, conduit, and booster pump, are significantly impaired in patients with arterial hypertension (AH) compared to healthy controls [76]. This impairment highlights the potential for LA dysfunction to occur before noticeable structural changes, emphasizing the importance of early assessment in identifying atrial abnormalities. Impaired function of the LA, particularly in reservoir and booster pump strain measured by the global longitudinal strain (GLS), strongly correlates with elevated NT-proBNP levels [75,77]. Additionally, LA reservoir and booster pump strain also show significant correlations in hypertensive patients [78]. This suggests that LA strain, as assessed by GLS, could serve as an early marker of LA dysfunction, potentially identifying issues before LA enlargement becomes apparent and enabling the earlier detection of HHD. However, more research is needed to establish definitive cut-off values for LA strain and validate its clinical utility.

## 11. Left Ventricular Hypertrophy (LVH)

LVH, a common manifestation of HHD, is characterized by left ventricular thickening (≥12 mm) or an increased left ventricular mass (LVM). It serves as a critical risk factor in patients both with and without cardiovascular disease, making its detection vital for accurate cardiovascular risk stratification and management.

Studies have demonstrated a significant correlation between NT-proBNP levels and LVM, suggesting that NT-proBNP may serve as a reliable biomarker in the detection of LVH [65,66,79]. The high negative predictive value indicates NT-proBNP’s potential utility in ruling out LVH in asymptomatic hypertensive patients [79]. This could, in turn, reduce the need for further diagnostic testing. Another study demonstrated a significant increase in NT-proBNP levels following exercise, with the largest rise observed in a subgroup of hypertensive patients with left ventricular remodeling [67]. This finding suggests that these patients are at a higher risk for developing left ventricular dysfunction. Such an approach could help identify high-risk patients who may benefit from more aggressive hypertension management, potentially preventing the progression to heart failure. However, these studies are relatively small, and further research is needed to generalize their outcomes and establish their utility in clinical practice. Additionally, well-defined cut-off values for NT-proBNP are required in order to enhance its diagnostic accuracy. While NT-proBNP shows greater diagnostic accuracy compared to transthoracic echocardiography in screening for LVH, it cannot fully replace it.

It is also important to note that patients with HFpEF may present with low NT-proBNP levels despite elevated wedge pressures [80]. This occurs because the smaller left ventricular dimensions and greater wall thickness reduce wall stress, leading to a lower NT-proBNP release. This phenomenon complicates the diagnosis of LVH and HFpEF, as the expected biomarker elevation may not be as pronounced, potentially leading to an underestimation of the severity of the condition. While NT-proBNP’s diagnostic utility is well-established in acute decompensated heart failure, it is less reliable in early-stage or treated HFpEF, highlighting the need for careful interpretation in these cases [80,81,82].

## 12. Systolic and Diastolic Dysfunction

HHD may eventually progress towards heart failure, where the initial myocardial hypertrophy and fibrosis result in myocardial stiffening and systolic as well as diastolic LV dysfunction. Eccentric hypertrophy is more likely to lead to HFrEF, while concentric hypertrophy may lead to HFpEF [10].

A significant correlation has been observed between NT-proBNP levels and the E/e’ ratio. Specifically, patients with higher NT-proBNP levels tend to have higher E/e’ ratios, which reflects worse diastolic dysfunction. This supports the understanding that an elevated NT-proBNP indicates a worse heart failure status, often associated with increased diastolic pressures and dysfunction [70]. However, in patients with an E/e’ ratio < 13, research has shown that parameters related to systolic function, such as the systolic tissue Doppler velocity (S’), are more closely associated with NT-proBNP levels. This suggests a reversed correlation in these cases: lower S’ values are associated with higher NT-proBNP levels. Thus, when LV filling pressures are not as elevated, subtle impairments in systolic function, rather than left atrial enlargement, may more significantly drive elevated NT-proBNP levels [72].

## 13. Conclusions

HHD represents a growing global health challenge due to the increasing prevalence of hypertension, particularly in low- and middle-income countries. Efforts to reduce the burden require coordinated strategies focusing on prevention, early diagnosis, and effective management, along with addressing broader social determinants of health. Hypertension is the most common modifiable risk factor for early cardiovascular disease and death, making continuous monitoring essential in order to detect complications and slow their progression. Persistent high blood pressure leads to left ventricular hypertrophy, which can eventually result in heart failure. The effective management of heart failure requires the early identification and correction of the underlying risk factors.

NT-proBNP has emerged as a pivotal biomarker for diagnosing and assessing the prognosis of heart failure. It plays an essential role in clinical decision-making, especially in the diagnosis, management, and prognosis of heart failure. Guidelines highlight the importance of considering individual patient factors when interpreting NT-proBNP levels, advocating for a personalized approach in managing heart failure. Significant advancements in the detection methods of NT-proBNP have enhanced its sensitivity, specificity, and usability in clinical settings. Looking ahead, the development of algorithms that integrate clinical and echocardiographic parameters along with NT-proBNP measurements is expected. This approach aims to more accurately diagnose and categorize patients, particularly those with a worsening prognosis, ensuring targeted and effective management strategies (Table 3).

## Figures and Tables

**Table 1 diagnostics-15-00113-t001:** Key points.

Hypertensive Heart Disease	A major cardiac complication of hypertension, leading to structural and functional changes in the heart, such as left ventricular hypertrophy, diastolic dysfunction, and, eventually, systolic dysfunction.
NT-proBNP	A promising diagnostic marker for HHD. Elevated levels can indicate cardiac stress and dysfunction.
Diagnosis of HHD	Early diagnosis and treatment are crucial for preventing progression to congestive heart failure. NT-proBNP can aid in early detection.
NT-proBNP Levels and Heart Failure	Heart failure likely if BNP > 400 pg/mL and NT-proBNP > 2000 pg/mL; and unlikely if BNP < 100 pg/mL and NT-proBNP < 400 pg/mL. Uncertainty exists between these ranges.
Monitoring NT-proBNP	Tracking NT-proBNP levels over time can assess treatment effectiveness and disease progression. Decreasing levels suggest a positive response to treatment.
Individualized Interpretation	Interpreting NT-proBNP levels should consider individual patient factors like age, renal function, and comorbidities.
Integration into Algorithms	Integrating NT-proBNP into diagnostic and therapeutic algorithms for cardiovascular diseases improves clinical decision-making and patient outcomes.

**Table 2 diagnostics-15-00113-t002:** Echocardiography findings in HHD and NT-proBNP levels.

Echocardiography	Increase of NT-proBNP	References
Left Ventricular Hypertrophy	NT-proBNP levels are often elevated in patients with LVH, reflecting the increased cardiac workload and stress. Importantly, NT-proBNP may be elevated even before LVH becomes apparent on echocardiography, suggesting its potential for early detection.	[65,66,67]
Diastolic Dysfunction	Characterized by impaired relaxation and filling of the left ventricle. Echocardiographic parameters like the E/e’ ratio assess diastolic function. NT-proBNP levels correlate with the severity of diastolic dysfunction.	[68,69]
Systolic Dysfunction	As HHD progresses, systolic dysfunction can develop, indicated by a reduced ejection fraction on echocardiography. NT-proBNP levels tend to rise with worsening systolic function, reflecting the declining pumping capacity of the heart.	[70]
Left Atrial Enlargement/remodeling	Chronic pressure overload in HHD can lead to LAE, visible on echocardiography. Elevated NT-proBNP levels are associated with LAE, indicating the impact of HHD on the left atrium.	[68,71,72]

**Table 3 diagnostics-15-00113-t003:** NT-proBNP plays a crucial role in the diagnosis of hypertensive heart disease.

Early Detection	Even before traditional markers like left ventricular hypertrophy become apparent, NT-proBNP can signal early microstructural changes in the left atrium, such as fibrosis. This early detection is vital for timely intervention
Correlation with Cardiac Stress	Elevated NT-proBNP levels correlate with increased cardiac stress, reflecting the strain placed on the heart due to hypertension.
Assessment of Dysfunction	NT-proBNP is valuable in assessing both diastolic and systolic dysfunction, key features of HHD. This helps identify atrial and ventricular impairment even before significant structural changes are visible.
Integration with Other Diagnostic Tools	While NT-proBNP is a powerful tool, it is most effective when used in conjunction with other clinical and echocardiographic assessments. This comprehensive approach ensures a more accurate diagnosis and risk stratification.
No Strict Cut-Offs	While general guidelines exist (e.g., CHF likely if NT-proBNP > 2000 pg/mL), there are no strict cut-off values for diagnosing HHD solely based on NT-proBNP. Interpretation should consider individual patient characteristics.

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
