# Peer review of "The Role of NT-proBNP Levels in the Diagnosis of Hypertensive Heart Disease"

_diagnostics, 2025, doi:10.3390/diagnostics15010113_

Round 1
Reviewer 1 Report
Comments and Suggestions for Authors
This manuscript revealed an association between NT-pro BNP and cardiovascular disease including HHD. These data send important messages. However, I have some questions. Please make clear. 
1. Although the usefulness of NT-pro BNP has been shown in comparison with BNP, the comparison is insufficient. Please clarify the differences in the evaluation of these, especially in HHD.
2. I agree that NT-pro BNP is useful for the evaluation of heart failure stage B. Is there a standard value for NT-pro BNP that can predict progression from heart failure stage A to B of?
3. Please describe the effect of treatment such as ARNI on BNP or NT-pro BNP.

Author Response
Comments 1: Although the usefulness of NT-pro BNP has been shown in comparison with BNP, the comparison is insufficient. Please clarify the differences in the evaluation of these, especially in HHD. |
Response 1: Thank you for pointing this out. We agree with this comment. Therefore, we have added a paragraph on page 5, lines 192-196.
|
Comments 2: I agree that NT-pro BNP is useful for the evaluation of heart failure stage B. Is there a standard value for NT-pro BNP that can predict progression from heart failure stage A to B of? |
Response 2: We have, added that although it would have been important and extremely useful to have cut off points, we don’t have standard values predicting progression from heart failure stage A to B. Page 7 lines 284-288. |
Reviewer 2 Report
Comments and Suggestions for Authors
Dear authors,
Although the paper is a review of already well-known facts, I believe that it is clearly and comprehensively written, and that it is interesting and useful for readers in a clinical sense.
Also, even it is very similarly conceived and written as the recently published review The Role of NT-proBNP Levels in the Diagnosis and Treatment of Heart Failure with Preserved Ejection Fraction—It Is Not Always a Hide-and-Seek Game, of course the mentioned paper is cited- reference 80, this review has its own specifics.
It is good that the importance of timely treatment of hypertension, as well as its role in the development of HFpEF first of all (the incidence of which is increasing), and then HFrEF, is emphasized, and in this light the similarity with the aforementioned research is justified, as well as the importance of the topic itself.
I would suggest minor changes:
Table 1. HHD - State the full name before the abbreviation or put the abbreviation in brackets where the name is mentioned for the first time
Line 43: Delete the empty space after the open bracket
Line 138: Instead of the word overt, I would suggest evident or manifest
After line 238: It would be useful to add a section on NT-proBNP in atrial fibrillation
Line 323: To correct (accept) what has already been corrected
Line 396: To correct (accept) what has already been corrected
Line 439: To correct (accept) what has already been corrected
Author Response
We truly appreciate for taking the time to review this manuscript. Please find the detailed corrections in red on the re-submitted files. We have corrected all the "suggested minor changes"